# Consultation patterns and frequent attenders in UK primary care from 2000 to 2019: a retrospective cohort analysis of consultation events across 845 general practices

Evangelos Kontopantelis ,[1,2] Maria Panagioti,[3,4] Tracey Farragher,[3] Luke A Munford ,[3] Rosa Parisi,[1,2] Claire Planner,[2,3,4] Sharon Spooner ,[3,5] Alice Tse,[3] Darren M Ashcroft,[4,6] Aneez Esmail[2,3]

► Prepublication history and additional supplemental material for this article are available online. To view these files, please visit the journal online (http://dx.doi.org/10.1136/bmjopen-2021-054666).

For numbered affiliations see end of article.

**Correspondence to**
Professor Evangelos Kontopantelis;
e.kontopantelis@manchester.ac.uk

## ABSTRACT

**Objective** To describe the distribution of consultations at the practice level and examine whether increases are uniform or driven by people who consult more frequently.

**Design** Retrospective cohort study.

**Setting** UK general practice data from the Clinical Practice Research Datalink (CPRD) GOLD database.

**Participants** 1 699 709 314 consultation events from 12 330 545 patients, in 845 general practices (1 April 2000 to 31 March 2019).

**Methods** Consultation information was aggregated by financial year into: all consultations/all staff; all consultations/general practitioners (GPs); face-to-face consultations/all staff; face-to-face consultations/GPs. Patients with a number of consultations above the 90th centile, within each year, were classified as frequent attenders. Negative binomial regressions examined the association between available practice characteristics and consultation distribution.

**Results** Among frequent attenders, all consultations by GPs increased from a median (25th and 75th centile) of 13 (10 and 16) to 21 (18 and 25) and all consultations by all staff increased from 27 (23–30) to 60 (51–69) over the study period. Approximately four out of ten consultations of any type concerned frequent attenders and the proportion of consultations attributed to them increased over time, particularly for face-to-face consultations with GPs, from a median of 38.0% (35.9%–40.3%) in 2000–2001 to 43.0% (40.6%–46.4%) in 2018–2019. Regression analyses indicated decreasing trends over time for face-to-face consultations and increasing trends for all consultation types, for both GPs and all staff. Frequent attenders consulted approximately five times more than the rest of the practice population, on average, with adjusted incidence rate ratios ranging between 4.992 (95% CI 4.917 to 5.068) for face-to-face consultations with all staff and 5.603 (95% CI 5.560 to 5.647) for all consultations with GPs.

**Conclusions** Frequent attenders progressively contributed to increased workload in general practices across the UK from 2000 to 2019. Important knowledge gaps remain in terms of the demographic, social and health

### Strengths and limitations of this study

► This is a large observational study that analysed 1 699 709 314 consultation events from 12 330 545 patients, in 845 general practices, from 1 April 2000 to 31 March 2019.

► The unique contribution of our study is the investigation of distribution trends among the top 10% of consulters for all consultations (covering all staff working in general practice), all consultations with GPs (including face-to-face consultations and all other consultations), face-to-face consultations with all staff, and face-to-face consultations with GPs.

► This is an observational study, using a database of electronic health records, and it is dependent on the consultations and consultation types being recorded accurately; we were not able to ascertain information on patient and practice characteristics which is critical to understanding why this group of frequent attending patients consult their GPs.

► This is a practice-level analysis that did not focus on patients, and future work should examine persistent frequent attendance, and the patient and practice characteristics associated with frequent attendance.

characteristics of frequent attenders and how UK general practices can be prepared to meet the needs of these patients.

## INTRODUCTION

General practice in the United Kingdom (UK) has been facing significant workload challenges for more than a decade. In a seminal paper published in 2016, Hobbs and colleagues showed that the crude annual consultation rate per person increased by 10.51%, from 4.67 in 2007–2008 to 5.16 in 2013–2014.[1] Using data covering nearly 100 million consultations from anonymised practices contributing data to the Clinical

Practice Research Datalink (CPRD) GOLD database, their study found that there was also a substantial increase in average consultation duration and total patient-facing clinical workload (by 16%) during this period. The authors concluded that the perceptions of a rapidly rising workload in UK general practice was well founded and compounded by the complexity of care that general practitioners (GPs) were having to provide including, for example, an increasing elderly population with multiple comorbidities. The 2016 report by the King's Fund also confirmed that UK general practice is in crisis with a workload that has increased significantly over time. In addition to indicating that the increase was due to the needs of an ageing population and more people with complex conditions and multimorbidity, it was noted that the effect was compounded by initiatives to transfer care from hospitals to the community.[2]

Although existing work supports the view that GP workload is increasing over time,[1] up-to-date research evidence on workload rates is lacking. One important omission in previous estimates of UK GP workload is that the distribution of consultations within the patient population of general practices has not been assessed. In the Netherlands, Smits and colleagues identified a group of patients, referred to as persistent frequent attenders, occupying the top 10% of age- and gender-adjusted attenders over three continuous years. This group of frequent attenders may have special health needs, and may require different management in practice due to their distinct clinical and psychological characteristics.[3 4] As most of the literature on frequent attenders has been produced in the Netherlands, the generalisability of these findings and potential solutions in the UK needs to be evaluated. Pilot work in the UK supports these findings and could mean that the top 10% of attenders in the UK general practice are responsible for utilising between 30% and 50% of all GP consultations.[5] However, there is a paucity of large studies in the UK to establish firm estimates of the prevalence of frequent attendance in the GP workload.

Moreover, working arrangements within UK general practices change rapidly and dynamically. Major reforms are being implemented in response to the 2016 General Practice Forward View which could have an impact on workload rates.[6] For example, tele(e)-consultation options with GPs and other staff members of general practices have become increasingly prevalent, reaching a peak during the COVID-19 pandemic.[7 8] The workforce composition of general practices has also changed as more allied health professionals (eg, advanced nurse practitioners, clinical pharmacists, and physician associates) and non-clinical staff have been appointed to increase the skill-mix in general practice and better serve patient needs.[9 10] In addition, the health needs of local populations may affect consultation rates and workload demands differently in subregions within the UK.[11 12] Thus, potential variations in the GP workload across geographic regions need to be identified to tailor potential solutions.

This study provides new information about this under-researched topic by analysing the distribution of consultations within general practices and practice populations, with a focus on frequent attenders. More specifically we examined: (a) different ways to describe consultation distributions in a way that is informative to clinical practice, dichotomising into high and low attendance groups; (b) how these vary across practices and regions, by consultation type; and (c) if and how they have changed over time. We also provide contemporary information about consultation rates to update the previous similar analysis undertaken 5 years ago.[1]

## METHODS
### Data
We conducted a retrospective cohort study using the CPRD GOLD database, which holds anonymised clinical information on patients, from participating general practices, including consultations, diagnoses, treatments, testing and referrals. Consultations include all contacts with the practice and will include requests for repeat prescriptions. They will also include consultations with practice staff (practice nurses, healthcare assistants, social prescribers, nurse practitioners and other allied health professional who are increasingly being employed by GPs). Some patients will contact a practice to speak to reception staff and these are usually identified as administrative episodes so will not be counted as consultations. Non face-to-face consultations will have included telephone and online consultations.

We analysed consultations data from 1 April 2000 until 31 March 2019, in annual bins of financial years. The number of practices in CPRD GOLD varied from 407 in 2000–2001, to 740 in 2008–2009, to 389 in 2018–2019. Of these, only 113 practices contributed data throughout the whole of the study period. Although the database is representative of the UK in terms of age, gender and deprivation,[13] it is only broadly geographically representative since it collects data from practices using the Vision clinical system, and clinical system usage is geographically clustered in the UK.[14] For example, Scottish general practices are more heavily represented in more recent years of the CPRD database. Socioeconomic deprivation of the practice location was available, as measured by the 2015 English Index of Multiple Deprivation (IMD), which is a composite score across seven domains: Income, Employment, Education, Skills and Training, Health and Disability, Crime, Barriers to Housing Services, Living Environment.[15] The Scottish, Welsh and Northern Irish IMDs are very similar, but with small domain changes to capture different country dynamics. Consultation information was aggregated within each financial year, for each active patient (registered for at least 1 day during the respective year). We aggregated the information into four consultation types: (1) all consultation types (including administrative) by all practice staff (health worker or administrator); (2) all consultation types by GPs; (3)

face-to-face consultations by all staff; and (4) face-to-face consultations by all GPs (online supplemental tables 1 and 2).

## Statistical analyses

The distribution of consultations within each practice was quantified using various approaches to allow for within and between practice comparisons over time. First, we calculated the mean, 10th, 25th, 50th (median), 75th and 90th centile of each of the four consultation types, for each practice within each financial year. Next we used the 50th, 75th and 90th centile to aggregate the consultation volume into dichotomous patient groups: those with a number of consultations in a given year below the 50th (or 75th or 90th) centile and those with a number of consultations equal to or higher than the 50th (or 75th or 90th) centile. We report the percentage of consultations attributed to the top group (equal to or above the 50th, 75th or 90th centile), within each year. Finally, to allow for comparisons across practices, we used an arbitrary threshold of 12 consultations within a year to dichotomise volume, across each consultation type. As for centiles, we report the percentage of practice consultations attributed to people with 12 or more, within each year. The reported percentages, for centiles and 12 consultations, and for each practice within a year, can be expressed as:

$$p_{ij} = 100 * con_{ij1} / (con_{ij1} + con_{ij2})$$

where:

$i = 1, \ldots, 4$ the consultation type

$j$ denotes the 50th, 75th, 90th centile or 12 consultations;

$con_{ij1}$ is the total consultation volume for the top group;

$con_{ij2}$ is the total consultation volume for the bottom group.

Since the groups were not always balanced as per the centiles (eg, dichotomising along the 50th centile did not result in two equal groups of patients), due to the relative small numbers of consultations for the average patient, we also calculated the ratio of consultations of the top group over the bottom group. This can be expressed as:

$$r_{ij} = (con_{ij1} / pat_{ij1}) / (con_{ij2} / pat_{ij2})$$

where:

$pat_{ij1}$ is the number of patients in the top group;

$pat_{ij2}$ is the number of patients in the bottom group.

We report these metrics in violin plots,[16] across all practices and by region, over time. Violin plots can be interpreted as box plots (including the median as a marker and a box indicating the interquartile range (IQR)), overlaid with the density of the distribution for better visualisation.

We used negative binomial regression models for key outcomes of interest (50th, 75th, 90th centile and 12 consultations) to examine the association between available practice characteristics and imbalanced consultation

**Table 1** Medians (25th–75th centiles) of average number of consultations per patient, by consultation type, all practices

| Year | All consultations by all staff | All consultations by GPs | Face-to-face consultations by all staff | Face-to-face consultations by GPs |
|---|---|---|---|---|
| 2000–2001 | 11.4 (9.8–13.0) | 5.3 (4.3–6.8) | 5.2 (4.5–6.3) | 3.7 (3.2–4.7) |
| 2001–2002 | 12.5 (10.5–14.2) | 5.3 (4.4–6.4) | 5.2 (4.5–6.4) | 3.6 (3.1–4.4) |
| 2002–2003 | 13.4 (11.2–15.7) | 5.3 (4.3–6.6) | 5.1 (4.3–6.4) | 3.5 (2.9–4.3) |
| 2003–2004 | 14.8 (12.0–17.3) | 5.7 (4.6–6.9) | 5.1 (4.3–6.8) | 3.4 (2.8–4.2) |
| 2004–2005 | 15.8 (12.5–18.3) | 5.9 (4.9–7.1) | 5.0 (4.1–6.7) | 3.2 (2.7–4.2) |
| 2005–2006 | 17.0 (13.8–20.1) | 6.3 (5.4–7.6) | 5.3 (4.4–7.0) | 3.3 (2.8–4.3) |
| 2006–2007 | 17.3 (14.0–20.5) | 6.3 (5.3–7.7) | 5.1 (4.2–6.9) | 3.2 (2.7–4.1) |
| 2007–2008 | 18.4 (15.3–21.2) | 6.5 (5.4–7.7) | 5.2 (4.3–6.6) | 3.3 (2.8–4.1) |
| 2008–2009 | 19.4 (16.5–22.4) | 6.8 (5.6–7.9) | 5.2 (4.4–6.4) | 3.3 (2.8–3.9) |
| 2009–2010 | 20.3 (17.7–23.3) | 7.1 (5.9–8.4) | 5.3 (4.5–6.4) | 3.4 (2.8–4.0) |
| 2010–2011 | 20.4 (17.8–23.8) | 7.3 (6.1–8.4) | 5.3 (4.4–6.3) | 3.4 (2.8–4.0) |
| 2011–2012 | 21.3 (18.5–24.6) | 7.6 (6.4–8.8) | 5.4 (4.5–6.4) | 3.4 (2.8–4.0) |
| 2012–2013 | 21.8 (19.0–25.2) | 7.6 (6.6–9.1) | 5.4 (4.6–6.4) | 3.4 (2.9–4.1) |
| 2013–2014 | 22.8 (19.7–26.3) | 8.0 (6.9–9.5) | 5.4 (4.6–6.4) | 3.4 (2.8–4.1) |
| 2014–2015 | 23.4 (20.3–26.7) | 8.2 (6.9–9.7) | 5.4 (4.6–6.3) | 3.3 (2.8–4.1) |
| 2015–2016 | 23.2 (20.0–26.8) | 7.9 (6.6–9.3) | 5.1 (4.3–6.1) | 3.2 (2.6–3.9) |
| 2016–2017 | 23.4 (20.3–27.4) | 8.3 (6.8–9.7) | 5.2 (4.3–6.2) | 3.3 (2.6–4.0) |
| 2017–2018 | 23.8 (20.7–27.4) | 8.2 (7.0–9.7) | 5.1 (4.1–6.1) | 3.2 (2.5–4.0) |
| 2018–2019 | 25.1 (21.7–28.7) | 8.3 (7.0–9.7) | 5.0 (4.2–6.0) | 3.1 (2.5–3.9) |

GP, general practitioner.

**Table 2** Medians (25th–75th centiles) of 90th centile number of consultations per patient, by consultation type, all practices

| Year | All consultations by all staff | All consultations by GPs | Face-to-face consultations by all staff | Face-to-face consultations by GPs |
|---|---|---|---|---|
| 2000–2001 | 27 (23–30) | 13 (10–16) | 12 (10–15) | 9 (8–11) |
| 2001–2002 | 30 (25–34) | 13 (10–16) | 12 (10–15) | 8 (7–11) |
| 2002–2003 | 32 (26–37) | 13 (10–16) | 12 (10–15) | 8 (7–10) |
| 2003–2004 | 35 (29–41) | 14 (12–17) | 12 (10–16) | 8 (7–10) |
| 2004–2005 | 38 (30–45) | 15 (12–18) | 12 (10–16) | 8 (7–10) |
| 2005–2006 | 41 (33–49) | 16 (13–19) | 12 (10–17) | 8 (7–10) |
| 2006–2007 | 42 (34–50) | 16 (13–19) | 12 (10–16) | 8 (7–10) |
| 2007–2008 | 44 (37–52) | 16 (13–19) | 12 (10–15) | 8 (7–10) |
| 2008–2009 | 47 (40–54) | 17 (14–20) | 12 (10–15) | 8 (7–10) |
| 2009–2010 | 49 (42–56) | 18 (15–21) | 12 (11–15) | 8 (7–10) |
| 2010–2011 | 49 (43–57) | 18 (15–21) | 12 (10–15) | 8 (7–10) |
| 2011–2012 | 51 (44–58) | 19 (16–22) | 13 (11–15) | 8 (7–10) |
| 2012–2013 | 52 (45–60) | 19 (16–23) | 13 (11–15) | 8 (7–10) |
| 2013–2014 | 55 (47–63) | 20 (17–24) | 13 (11–15) | 8 (7–10) |
| 2014–2015 | 56 (49–64) | 20 (17–24) | 13 (11–15) | 8 (7–10) |
| 2015–2016 | 55 (48–64) | 20 (17–23) | 13 (11–15) | 8 (7–10) |
| 2016–2017 | 56.5 (48–66) | 21 (17–24) | 12.5 (10–15) | 8 (7–10) |
| 2017–2018 | 57 (50–66) | 20 (17–24) | 12 (10–15) | 8 (6–10) |
| 2018–2019 | 60 (51–69) | 21 (18–25) | 12 (10–14) | 8 (6–10) |

GP, general practitioner.

distribution, with a focus on frequent attendance. The outcome was the average number of consultations per patient within each of the two groups, for each practice within a year, adjusted for the number of patients within each group (exposure term). The binary group predictor (top/bottom group) was interacted with each of the following practice characteristics, in separate models: list size (time varying), region, and practice location deprivation as measured by the 2015 IMD.[15]

All analyses were conducted with Stata v16 and an alpha level of 5% was used for the inferential statistics. We focused on the 90th centile analyses in the main article, splitting the population within a practice to the top 10% in terms of service usage within a year, and contrasting it to the remaining 90% of the practice population. As a sensitivity, we analysed data only from the 113 practices that were active throughout the whole of the study period.

## PATIENT AND PUBLIC INVOLVEMENT
The research was discussed with a Patient and Public Involvement (PPI) group (PRIMER, based at the Centre for Primary Care and Health Service Research at the University of Manchester) at a meeting held prior to analysis. The Group agreed that understanding consultation patterns was an important starting point for exploring the reasons for frequent attendance (eg, availability of double appointments for patients with multimorbidity),

the extent to which the needs of frequent attenders are being met, and their experience of primary care services.

## RESULTS
Overall, 1 699 709 314 consultation events across 12 330 545 patients in 845 general practices were analysed. We present the 90th centile analyses in the main article, and results were consistent for the other dichotomies (median, 75th centile and 12 or more consultations in a year), with summary results presented in the Supplementary Material file (online supplemental tables 3 to 10). Over time, the average volume of all consultations for a single patient, with GPs and all staff, increased. However, face-to-face consultations remained relatively stable and face-to-face consultations with GPs even decreased (table 1).

Over time, face-to-face consultations per annum with GPs and all staff remained stable for the top 10% group, and in 2018–2019 medians (25th and 75th centiles) were 8 (6 and 10) and 12 (10 and 14), respectively (table 2). Conversely, medians for all consultations with GPs and all staff greatly increased over time. All consultations with GPs increased from a median (25th and 75th centiles) of 13 (10 and 16) in 2000–2001 to 21 (18 and 25) in 2018–2019. Similarly, all consultations with all staff increased from 27 (23 and 30) in 2000–2001 to 60 (51 and 69) over the study period. However, the picture was somewhat

**Table 3** Medians (25th–75th centiles) of proportion of high volume (≥90th centile) per patient, by consultation type, all practices

| Year | All consultations by all staff | All consultations by GPs | Face-to-face consultations by all staff | Face-to-face consultations by GPs |
|---|---|---|---|---|
| 2000–2001 | 35.2 (33.9–36.5) | 38.0 (35.7–40.1) | 35.8 (34.4–37.7) | 38.0 (35.9–40.3) |
| 2001–2002 | 35.5 (34.1–36.8) | 38.4 (36.5–40.4) | 36.3 (34.6–38.2) | 38.5 (36.5–40.9) |
| 2002–2003 | 35.6 (34.3–37.0) | 38.7 (36.6–41.1) | 36.5 (35.0–38.4) | 38.5 (36.4–41.2) |
| 2003–2004 | 35.9 (34.5–37.2) | 39.1 (37.2–41.5) | 36.9 (35.2–39.0) | 39.0 (36.7–41.7) |
| 2004–2005 | 36.5 (35.0–37.9) | 40.1 (37.8–42.4) | 37.9 (35.9–40.3) | 40.1 (37.6–43.3) |
| 2005–2006 | 36.1 (34.9–37.6) | 39.8 (37.6–41.9) | 37.5 (35.7–39.7) | 39.8 (37.3–42.5) |
| 2006–2007 | 36.7 (35.2–38.1) | 40.5 (38.2–42.8) | 38.3 (36.3–40.7) | 40.7 (38.2–44.2) |
| 2007–2008 | 36.5 (35.1–37.9) | 40.0 (38.2–42.1) | 37.9 (36.1–40.0) | 39.9 (37.9–42.9) |
| 2008–2009 | 36.6 (35.3–37.9) | 40.2 (38.3–42.1) | 38.1 (36.5–40.0) | 40.1 (37.8–43.1) |
| 2009–2010 | 36.3 (35.1–37.7) | 40.0 (38.4–41.9) | 38.0 (36.3–40.0) | 40.3 (38.1–43.0) |
| 2010–2011 | 36.7 (35.3–38.0) | 40.1 (38.5–42.0) | 38.3 (36.6–40.5) | 40.6 (38.3–43.1) |
| 2011–2012 | 36.9 (35.5–38.1) | 40.4 (38.7–42.1) | 38.6 (37.0–40.6) | 40.9 (38.6–43.5) |
| 2012–2013 | 36.9 (35.5–38.2) | 40.3 (38.7–41.9) | 38.6 (37.1–40.5) | 40.9 (38.7–43.3) |
| 2013–2014 | 36.8 (35.5–38.3) | 40.5 (39.1–42.2) | 38.9 (37.2–40.7) | 41.3 (39.4–43.6) |
| 2014–2015 | 37.1 (35.6–38.5) | 40.8 (39.3–42.5) | 39.4 (37.5–41.4) | 41.5 (39.3–44.0) |
| 2015–2016 | 36.6 (35.3–38.2) | 41.4 (39.7–43.7) | 40.4 (38.4–42.7) | 43.1 (40.4–46.3) |
| 2016–2017 | 36.8 (35.6–38.4) | 41.0 (39.6–43.0) | 40.1 (38.1–42.4) | 42.5 (40.3–45.4) |
| 2017–2018 | 36.8 (35.3–38.3) | 40.9 (39.1–42.9) | 40.1 (38.1–42.2) | 42.8 (40.1–45.4) |
| 2018–2019 | 36.6 (35.3–38.0) | 40.9 (39.2–42.7) | 40.1 (38.3–42.5) | 43.0 (40.6–46.4) |

GP, general practitioner.

different when quantifying service usage in relative terms, with increases over time across all consultations types for the top 10% group (table 3 and figure 1). The percentage of face-to-face consultations attributed to the top 10% of users within a year increased from a median (25th and 75th centiles) of 38.0% (35.9%–40.3%) to 43.0% (40.6%–46.4%) for interactions with GPs, and from 35.8% (34.4%–37.7%) to 40.1% (38.3%–42.5%)

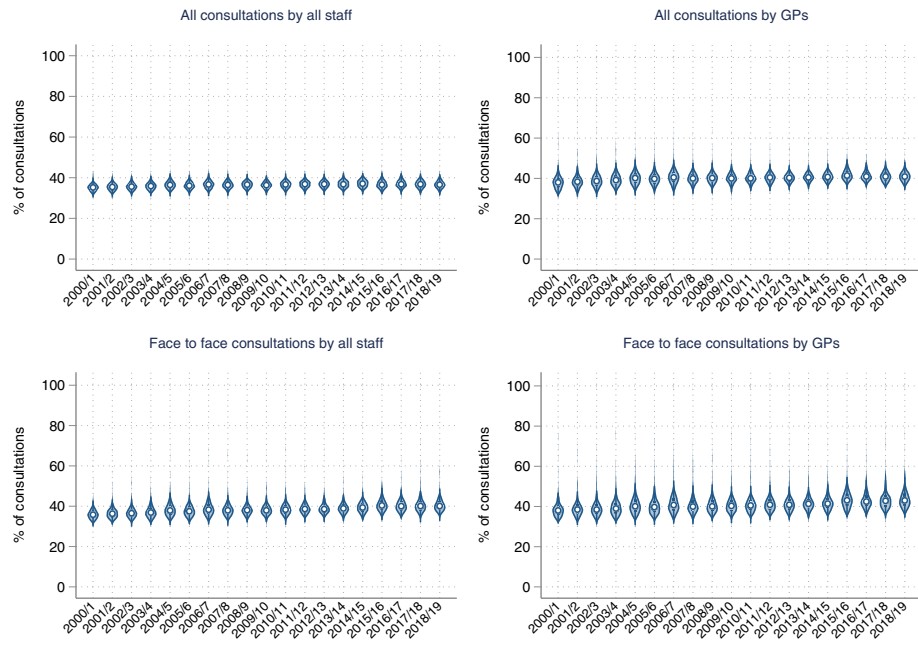

**Figure 1** Percentage of consultations by people consulting at the 90th centile or above within a financial year, over time. GP, general practitioner.

**Table 4** Medians (25th–75th centiles) of ratio for users ≥90th centile over the rest of consultations per patient, by consultation type, all practices

| Year | All consultations by all staff | All consultations by GPs | Face-to-face consultations by all staff | Face-to-face consultations by GPs |
|---|---|---|---|---|
| 2000–2001 | 4.6 (4.4–4.9) | 4.9 (4.6–5.4) | 4.5 (4.3–4.8) | 4.7 (4.5–5.1) |
| 2001–2002 | 4.7 (4.5–5.0) | 5.0 (4.7–5.5) | 4.6 (4.4–4.9) | 4.8 (4.5–5.2) |
| 2002–2003 | 4.7 (4.5–5.1) | 5.2 (4.8–5.6) | 4.7 (4.4–5.0) | 4.9 (4.6–5.3) |
| 2003–2004 | 4.8 (4.6–5.1) | 5.3 (5.0–5.8) | 4.8 (4.5–5.1) | 5.0 (4.7–5.5) |
| 2004–2005 | 5.0 (4.7–5.3) | 5.5 (5.1–6.1) | 5.0 (4.6–5.4) | 5.2 (4.8–5.8) |
| 2005–2006 | 4.9 (4.6–5.2) | 5.5 (5.1–6.0) | 4.8 (4.6–5.3) | 5.1 (4.8–5.7) |
| 2006–2007 | 5.0 (4.7–5.4) | 5.7 (5.2–6.3) | 5.1 (4.7–5.6) | 5.4 (4.9–6.0) |
| 2007–2008 | 5.0 (4.7–5.3) | 5.6 (5.1–6.0) | 5.0 (4.7–5.4) | 5.2 (4.8–5.7) |
| 2008–2009 | 5.0 (4.7–5.4) | 5.6 (5.2–6.0) | 5.0 (4.7–5.4) | 5.3 (4.9–5.7) |
| 2009–2010 | 5.0 (4.7–5.3) | 5.6 (5.2–6.0) | 5.0 (4.7–5.4) | 5.2 (4.9–5.7) |
| 2010–2011 | 5.1 (4.8–5.4) | 5.7 (5.3–6.1) | 5.1 (4.7–5.5) | 5.3 (4.9–5.8) |
| 2011–2012 | 5.1 (4.8–5.4) | 5.7 (5.3–6.1) | 5.1 (4.8–5.6) | 5.4 (5.0–5.8) |
| 2012–2013 | 5.1 (4.8–5.4) | 5.7 (5.4–6.1) | 5.2 (4.8–5.6) | 5.4 (5.0–5.8) |
| 2013–2014 | 5.1 (4.8–5.5) | 5.8 (5.4–6.2) | 5.2 (4.9–5.6) | 5.5 (5.1–6.0) |
| 2014–2015 | 5.2 (4.9–5.5) | 5.9 (5.5–6.3) | 5.3 (5.0–5.8) | 5.6 (5.2–6.1) |
| 2015–2016 | 5.1 (4.8–5.5) | 6.0 (5.6–6.5) | 5.6 (5.1–6.2) | 5.9 (5.4–6.6) |
| 2016–2017 | 5.1 (4.9–5.5) | 5.9 (5.5–6.4) | 5.5 (5.1–6.0) | 5.8 (5.3–6.4) |
| 2017–2018 | 5.1 (4.8–5.5) | 5.9 (5.5–6.3) | 5.5 (5.1–6.0) | 5.8 (5.4–6.5) |
| 2018–2019 | 5.1 (4.8–5.4) | 5.9 (5.5–6.3) | 5.5 (5.1–6.0) | 5.9 (5.4–6.6) |

GP, general practitioner.

for interactions with all staff. Percentage increases for all consultations were smaller, from a median (25th and 75th centiles) of 38.0 (35.7%–40.1%) to 40.9 (39.2%–42.7%) with GPs, and from 35.2 (33.9%–36.5%) to 36.6 (35.3%–38.0%) with all staff, over the study period. These increases in service usage for the top 10% group, across all four outcomes, were also reflected in the ratio of consultations, with the largest increase observed in face-to-face consultations with GPs, from a median (25th and 75th centiles) of 4.7 (4.5 and 5.1) in 2000–2001 to 5.9 (5.4 and 6.6) in 2018–2019 (table 4 and figure 2).

There was relatively little regional variability in the median percentage attribution of all four consultation categories, across all examined patient dichotomies. For example, the median percentage of face-to-face consultations with GPs for the top 10% of patients in terms of service usage with their respective practice, varied from 35% for Wales and West Midlands to 38% for Northern Ireland, over the whole study period.

Results from the negative binomial regression analyses indicated decreasing trends over time for face-to-face consultations and increasing trends for all consultation types, for both GPs and all staff (table 5 and figure 3). Adjusted incidence rate ratios (IRRs) for the top 10% consultations group ranged between 4.99 (95% CI 4.92 to 5.07) for face-to-face consultations with all staff, and 5.60 (95% CI 5.56 to 5.65) for all consultations with GPs,

compared with the bottom 90% consultations group. Associations between deprivation and consultations in the two groups were weak. Some regional variation was observed, with Scotland associated with the highest rate ratios for face-to-face consultations with GPs, and the South West associated with the highest rate ratios for face-to-face consultations with all staff. Pseudo-$R^2$ values for the four models varied between 0.048 for face-to-face consultations by all staff and 0.109 for all consultations by all staff.

Results from analyses of the other dichotomies (75–25, 50–50 and 12 or more consultations) and sensitivity analyses focusing on practices for the whole of the study period were broadly similar to our main results.

## DISCUSSION
### Summary of findings
This study examined distribution trends in four types of consultations (all consultations by GPs; all consultations by all staff; face-to-face consultations by GPs; face-to-face consultations by all staff) among frequent attenders (the top 10% of all consulters) and the rest of consulters in the UK general practices over the period from 2000–2001 to 2018–2019.

We found that all consultations by GPs and all consultations by all staff have considerably increased over this time

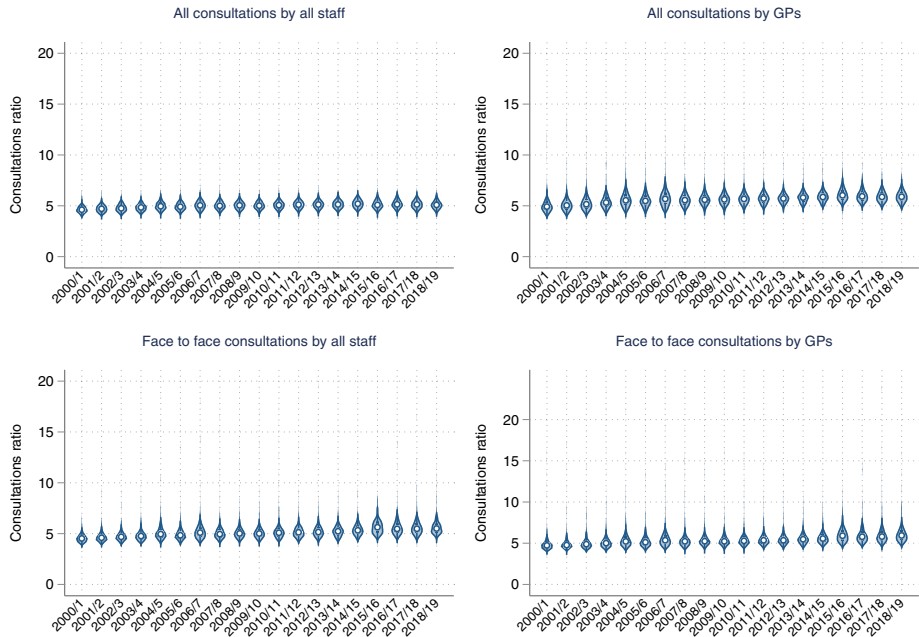

**Figure 2** Ratio of consultations for people ≥90th centile versus those below within a financial year, over time. GP, general practitioner.

period. Specifically, all consultations by GPs per person increased from a median of 5 to 8 and all consultations by all staff increased from 11 to 25. However, face-to-face consultations by GPs and face-to-face consultations by all staff have remained static (and may even have decreased).

Frequent attenders had five times more consultations of any type compared with the rest of consulters over the study period. Among frequent attenders, all consultations by GPs increased from a median of 13 to 21 and all consultations by all staff increased from 27 to 60 over the study period. Approximately four out of ten consultations of any type concerned frequent attenders and the proportion of consultations attributed to frequent attenders increased over time, particularly for face-to-face consultations with GPs (from 38% in 2000–2001 to 43% in 2018–2019). We found little evidence of regional variability in the attribution of all four consultation types, across all categories of consulters. The only exemption was that the distribution of face-to-face consultations with GPs was highest in Scotland and the distribution of face-to-face consultations with all staff was highest in Northern Ireland.

### Research in context

The unique contribution of our study is the investigation of distribution trends in all four types of consultations among frequent attenders (the top 10% of consulters). No previous study in the UK has examined the contribution that frequent attenders make to the workload of general practices. Our findings showed that frequent attenders account for an increasing proportion of face-to-face consultations with GPs (38%–43%) and are responsible for nearly 40% of all four types of consultations fairly

constantly over time. This striking finding suggests that a relatively small number of patients are accounting for a large proportion of GP workload including face-to-face consultations. This could be potentially attributed to a relative reduction in accessibility to face-to-face consultations for non-frequent attenders, which may be driven by a perception of some or all actors (patient/carer/clinician/practice staff) that the need for face-to-face contact is smaller in this patient group. Despite the importance of these findings, there is very little research on frequent attenders in the UK to make direct comparisons. Outside the UK, our findings are consistent with epidemiological studies conducted in the Netherlands which have also shown that frequent attenders were responsible for nearly 40% of the face-to-face consultations.[3] Similarly, a previous review on frequent attenders reported that the top 10% of attenders accounted for 30%–50% of all primary care consultations.[17]

Our current knowledge of the demographic, clinical and social characteristics of frequent attenders is scarce in the UK whereas findings from Europe are also inconsistent. Our analysis suggests that frequent attenders are identified across all parts of the UK and that deprivation, practice size or regional variation are not drivers for the number of frequent attenders by practice. There is evidence from Europe that frequent attenders (and particularly persistent frequent attenders) are more likely to be female and older, present more social and psychiatric problems, receive more prescriptions of psychotropic medication, have more medically unexplained physical symptoms, and more chronic medical conditions.[3 17] Our study cannot determine whether these

**Table 5** Associations between covariates of interest and consultation volume, incidence rate ratios (and 95% confidence intervals) from main negative binomial regression model

| Parameter | All consultations – all staff | All consultations – GPs | Face-to-face consultations – all staff | Face-to-face consultations – GPs |
|---|---|---|---|---|
| Year | 1.046 (1.045 to 1.047) | 1.028 (1.028 to 1.029) | 0.995 (0.994 to 0.997) | 0.988 (0.986 to 0.989) |
| List size (per 1000 patients) | 1.003 (1.002 to 1.004) | 0.997 (0.996 to 0.998) | 1.001 (0.998 to 1.003) | 0.992 (0.990 to 0.995) |
| Consultation group | | | | |
| Bottom 90% | Reference category | | | |
| Top 10% | 5.038 (5.007 to 5.068) | 5.603 (5.560 to 5.647) | 4.992 (4.917 to 5.068) | 5.253 (5.171 to 5.337) |
| IMD deprivation quintile | | | | |
| 1 (least deprived) | Reference category | | | |
| 2 | 1.005 (0.994 to 1.016) | 1.019 (1.005 to 1.033) | 0.993 (0.967 to 1.020) | 1.044 (1.015 to 1.073) |
| 3 | 1.048 (1.037 to 1.059) | 1.049 (1.035 to 1.063) | 1.038 (1.011 to 1.065) | 1.076 (1.047 to 1.106) |
| 4 | 1.019 (1.009 to 1.030) | 1.046 (1.032 to 1.060) | 0.981 (0.956 to 1.006) | 1.058 (1.030 to 1.087) |
| 5 (most deprived) | 1.025 (1.015 to 1.036) | 0.993 (0.980 to 1.006) | 0.997 (0.972 to 1.023) | 1.007 (0.981 to 1.035) |
| Region | | | | |
| North East | Reference category | | | |
| North West | 0.926 (0.899 to 0.953) | 0.995 (0.958 to 1.034) | 1.033 (0.961 to 1.111) | 0.985 (0.913 to 1.063) |
| Yorkshire and Humber | 0.952 (0.920 to 0.985) | 1.028 (0.984 to 1.075) | 1.023 (0.940 to 1.113) | 1.003 (0.918 to 1.096) |
| East Midlands | 0.898 (0.867 to 0.930) | 0.912 (0.872 to 0.955) | 0.963 (0.883 to 1.050) | 0.915 (0.835 to 1.002) |
| West Midlands | 0.840 (0.815 to 0.865) | 1.034 (0.994 to 1.074) | 1.074 (0.998 to 1.157) | 1.065 (0.986 to 1.150) |
| East of England | 0.853 (0.827 to 0.880) | 0.946 (0.909 to 0.985) | 1.012 (0.937 to 1.092) | 0.960 (0.886 to 1.040) |
| South West | 0.954 (0.926 to 0.983) | 1.085 (1.043 to 1.128) | 1.256 (1.166 to 1.353) | 1.080 (0.999 to 1.168) |
| South Central | 0.844 (0.819 to 0.870) | 0.952 (0.916 to 0.990) | 1.100 (1.021 to 1.186) | 1.052 (0.972 to 1.138) |
| London | 0.738 (0.716 to 0.760) | 0.922 (0.887 to 0.958) | 0.895 (0.832 to 0.964) | 0.918 (0.850 to 0.992) |
| South East Coast | 0.830 (0.805 to 0.855) | 0.928 (0.893 to 0.964) | 0.987 (0.918 to 1.063) | 1.000 (0.926 to 1.080) |
| Northern Ireland | 1.150 (1.115 to 1.187) | 1.446 (1.389 to 1.506) | 1.087 (1.006 to 1.174) | 1.128 (1.040 to 1.223) |
| Scotland | 0.785 (0.763 to 0.808) | 1.042 (1.004 to 1.081) | 1.249 (1.163 to 1.340) | 1.207 (1.121 to 1.300) |
| Wales | 0.883 (0.858 to 0.909) | 0.969 (0.934 to 1.006) | 1.045 (0.973 to 1.123) | 1.007 (0.934 to 1.086) |

GP, general practitioner; IMD, Index of Multiple Deprivation.

factors are the main driving force for frequent attendance but the contribution to overall GP workload needs further examination. A systematic review focused on children also found that frequent attendance in children was associated with presence of psychosocial and mental health problems, younger age, school absence, presence of chronic conditions, and high level of anxiety in their parents.[18] However, another systematic review on older adults found that frequent attendance in late life was mainly associated with presence and severity of chronic

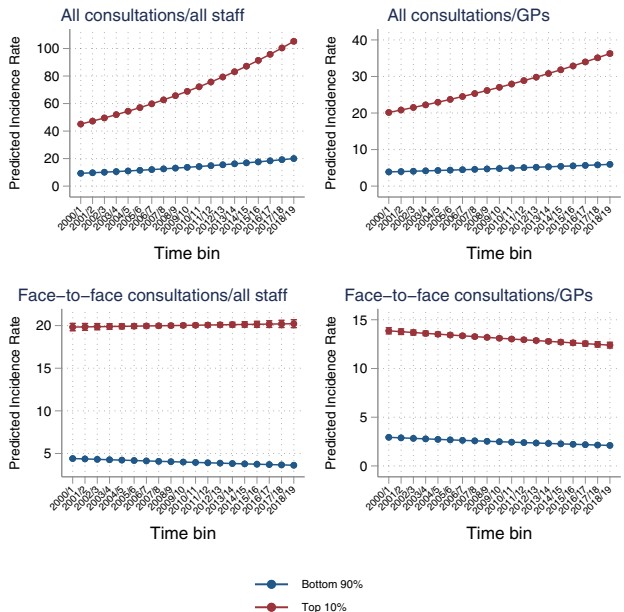

**Figure 3** Consultation count estimates by group over time (predicted incidence rates from the negative binomial regression model with a time-group interaction term).* *Interpreted as the estimated average number of consultations for a patient within each group. GP, general practitioner.

medical conditions whereas associations with mental conditions, drug use and social support were heterogeneous.[19] Thus, although there is evidence that frequent attenders differ in terms of demographic, social, mental and physical characteristics from the rest of consulters, their profile may vary across age groups and according to different definitions of frequent attendance (eg, persistent or non-persistent). Our findings and PPI work strongly support the need for future research to characterise frequent attenders in UK general practice taking into consideration the evidence and challenges encountered by previous studies conducted mostly in Europe.

Moreover, this study confirms and extends the secular trends first identified by Hobbs and colleagues that activity in general practice has increased over time.[1] This previous study found that the crude annual consultation rate per person increased from 4.67 in 2007–2008 to 5.16 in 2013–2014 and that GP telephone consultation rates doubled, compared with a 5.20% rise in face-to-face consultations. Consistent with this, we found that all consultations by GPs have increased and that face-to-face consultations are not the main driver of this increase. This finding suggests that other modalities for consulting such as telephone or online consultations may have become more important in recent years. Face-to-face consultations may have reached a capacity ceiling and other forms of consultations may be used to address otherwise unmet health needs. Moreover, Hobbs *et al*[1] found that the greatest increases in rates of consultations were in GPs, with a rise of 12.37% per 10 000 person-years, compared with 0.9% for practice

nurses. In our study, however, the highest increase over time was observed in all consultations by all staff with consultation medians increasing from 11 in 2000–2001 to 25 in 2018–2019. This impressive increase in all consultations by all staff may suggest that the increase in multidisciplinary staff working in general practices has resulted in a wider range of practice staff dealing with patients and that it is not all driven by face-to-face consultations. This trend appears to predate the 2016 General Practice Forward View and is likely to continue.

### Limitations
This study has a number of important limitations. First, we used our own definition of frequent attenders (top 10% of consulters) and therefore our findings are only broadly comparable with other studies which have used other definitions (ie, top 10% of consulters over 3 years; top 25% stratified by age, gender of health conditions). It is reassuring that our definition of frequent attendance is consistent with the recommended definition in a recent systematic review,[20] that our sensitivity analyses showed similar results, and that our findings were similar to the findings from our previous pilot study. Second, patients who have left a general practice but have not been removed from the register (also known as 'ghost' patients) may exaggerate the differences between top and bottom users, and we know that their numbers vary by region.[21] This is an observational study, using a database of electronic health records, and it is dependent on the consultations and consultation types being recorded accurately. Since UK primary care has been almost fully computerised by the start of this century,[22] it is extremely unlikely that any consultations fail to be recorded. However, the default setting in recording a consultation is face-to-face, and this needs to be changed by the health professional for other consultation types. Thus, face-to-face consultations may be overestimated, if some health professionals miss that step. In addition, the underlying workload across consultation types may differ systematically, on top of expected patient variation, but this is something we cannot capture within this study of practice aggregates. Although the CPRD GOLD database is representative in terms of deprivation and population characteristics,[13] it is collecting data from a single computer system (Vision) and the contributing practices are not uniformly distributed across English regions, while its market share is in decline.[14] Thus, generalisability to every English region is questionable, especially in relation to the sensitivity analyses where the sample is quite small and the distribution of the practices across regional and deprivation strata changes. IMD scores are standardised within each UK country, and between-country comparisons cannot capture baseline differences across countries. However, our regression analyses on the relative difference between high and low deprivation, in the context of the outcome of interest, should be robust to these nuances. Finally, we have not examined persistent frequent attendance,

or the patient or practice characteristics associated with frequent attendance.

## Implications for policy and practice

Frequent attenders appear to be a major driver for the increase in consultations that have contributed to perceptions of increased workload in general practice. While many of these patients may have comorbidities and may need to be seen regularly, research suggest that they have wider social and psychological needs.[17] GPs should be looking at this group of patients more closely to understand who are they and why are they consulting more frequently. Our PPI work indicates that patients see this as critically important for improving quality of care and preventing patients from being stigmatised for frequent attendance. One key challenge for future policy is how can general practice be prepared to provide services and support to frequent attenders without medicalising them but addressing the wider social and psychological problems that they may have.

The large increase in the general practice workload in the last 20 years means that having extended multidisciplinary teams to meet a wide range of patient needs through a range of ways (eg, remote consultations) is perhaps the only solution for sustaining a viable primary care. This change in the workforce composition of general practices is reflected in the activities that we report (medians for all consultations by GPs increased 5 to 8 whereas all consultations by all staff increased from 11 to 25) and echoes recent policy documents such as the 2016 General Practice Forward View.[6] It is also likely that a multidisciplinary workforce in general practice could also better address the wider, not always medical, needs of frequent attenders.

## CONCLUSIONS

This is the first study to show that frequent attenders, a relatively small group of consulters (the top 10%), largely and progressively contributed to increased workload in general practices across the UK over the last 20 years. Important knowledge gaps remain in terms of the demographic, social and health characteristics of frequent attenders in the UK and how general practices can be prepared to meet the needs of these patients. We intend to explore these factors in a separate study. Our findings depict a new model of work in general practice whereby an increasing number of consultations are conducted by other staff members (rather than GPs) using alternative means (rather than face-to-face consultations). With the overall workload of general practices continuously rising, multidisciplinary and technology-enabled general practices could better meet the multiple and wider needs of frequent attenders and the rest of consulters.

**Author affiliations**
[1]Division of Informatics, Imaging and Data Science, The University of Manchester, Manchester, UK
[2]National Institute for Health Research (NIHR) School for Primary Care Research, Oxford, UK
[3]Division of Population Health, Health Services Research & Primary Care, The University of Manchester, Manchester, UK
[4]NIHR Greater Manchester Patient Safety Translational Research Centre, The University of Manchester, Manchester, UK
[5]Health Organisation, Policy and Economics (HOPE) Group, Centre for Primary Care & Health Services Research, The University of Manchester, Manchester, UK
[6]Division of Pharmacy & Optometry, The University of Manchester, Manchester, UK

**Acknowledgements** The authors thank the members of the PRIMER patient and public involvement group (funded through the National Institute for Health Research (NIHR) School for Primary Care Research) for their input into the early stages of this research and Julie Billsborough, Manoj Mistry, Carole Bennett and Ailsa Donnelly for their contributions.

**Contributors** The research idea originated with AE and AT, but all authors designed the study. EK extracted the data and conducted the analyses, and LM and RP supported him in this. EK is the guarantor of the work and accepts full responsibility for the presented content. TF led the Independent Scientific Advisory Committee (ISAC) submission, for access to the Clinical Practice Research Datalink (CPRD) GOLD database. EK, MP and AE drafted the manuscript, and TF, LAM, RP, CP, SS, AT and DA critically revised it.

**Funding** The authors have not declared a specific grant for this research from any funding agency in the public, commercial or not-for-profit sectors.

**Competing interests** None declared.

**Patient consent for publication** Not applicable.

**Ethics approval** This study is based on data from the Clinical Practice Research Datalink (CPRD) GOLD database obtained under license from the Medicines and Healthcare products Regulatory Agency (MHRA). The data are provided by patients and collected by the National Health Service (NHS) as part of their care and support. The interpretation and conclusions contained in this study are those of the authors alone, and not necessarily those of the MHRA, the National Institute for Health Research (NIHR), NHS, or the Department of Health. Approval to conduct this study using the CPRD was granted by the Independent Scientific Advisory Committee (ISAC) of the MHRA (protocol 20_192R). The authors thank the contributing patients and practices to the CPRD GOLD database who have allowed their data to be used for research purposes.

**Provenance and peer review** Not commissioned; externally peer reviewed.

**Data availability statement** Data may be obtained from a third party and are not publicly available. Data can be obtained from the CPRD and are not freely available.

**ORCID iDs**
Evangelos Kontopantelis http://orcid.org/0000-0001-6450-5815
Luke A Munford http://orcid.org/0000-0003-4540-6744
Sharon Spooner http://orcid.org/0000-0001-6965-3673

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
