## [Reviewer comments · BMJ Open]

ARTICLE DETAILS

TITLE (PROVISIONAL)	Consultation patterns and frequent attenders in UK primary care from 2000 to 2019: a cohort analysis of consultation events across 845 general practices.
AUTHORS	Kontopantelis, Evangelos; Panagioti, Maria; Farragher, Tracey; Munford, Luke; Parisi, Rosa; Planner, Claire; Spooner, Sharon; Tse, Alice; Ashcroft, Darren; Esmail, Aneez

VERSION 1 – REVIEW

REVIEWER	Friedberg, Mark
REVIEW RETURNED	30-Jul-2021

GENERAL COMMENTS	Thank you for your responses and for making corresponding changes to the manuscript.
--

REVIEWER	Soley-Bori, Marina King's College London
REVIEW RETURNED	15-Aug-2021

GENERAL COMMENTS	General comment: This paper aims to quantify the extent to which frequent attenders drive primary care consultations. The analysis uses an impressively large dataset, CPRD GOLD, with 12m patients and 845 practices. Analyses are conducted at the practice level. The authors motivate the significance of this work by arguing that frequent attenders may have special health needs. The authors also point at the changes in the primary care workforce mix (e.g., more allied health professionals) and care delivery (e.g., more e-consultations). However, the findings from this paper are unable to inform these two areas. Without information on patient and practice characteristics, the significance and usability of this work is critically hindered in my opinion. I provide further comments below. The authors have tried to address the comments from previous reviewers and updated the text accordingly. The new version of the manuscript is clearer but, in my opinion, there are still some major concerns, which I detail below. Major points: 1. Alignment between study goals and methodology: a) The methods section in the abstract should mention the regression analysis. Isn't that the main method to identify the extent to which high users drive primary care use? Under the strengths and limitations section, the regression results are not presented, only the descriptive on the percentage of consultations used by frequent users.
--

	b) In my opinion, the statement of the study goals in the introduction needs to be more focused (p.6, line 3). Particularly “a) different ways to describe consultation distributions in a way that is informative to clinical practice” seems quite general and vague. I think the study would come across as stronger if it was focused on high users. c) On page 7, line 48, the study goal is stated as understanding practice characteristics that drive an imbalance consultation distribution. This analysis lacks a clear study hypothesis. Again, I think focusing the paper on high users is preferable. d) What is the rationale behind the choice of the independent variables in the regression model? Why was deprivation included but not, for example, the percentage of people above 70? or the percentage of women? or practice characteristics (such as available workforce?). Adjusting for at least age is essential to understand the impact of being a "high user", unconfounded by this key factor. e) What was the goodness of fit of the regression model? f) Are the measures presented on page 7 (pij and rij) used beyond descriptives? How many regression models were estimated? I'm under the impression that for each frequent user threshold, two regression models were calibrated, one for pij and another for rij? But then I don't understand how the averages for each group in number of consultations come in and how they help understand the distribution of primary care consultations. Please clarify. g) In my opinion, the results section should be much more focused on high users and provide a stronger interpretation for the regression results. h) The implications for policy do not seem to flow from the study results: “Whilst many of these patients may have comorbidities and may need to be seen regularly, research suggest that they have wider social and psychological needs(17). GPs should be looking at this group of patients more closely to understand who are they and why are they consulting more frequently.” The authors use a reference (17) rather than their study findings, which is odd. i) Wouldn't the authors need a model explaining the change in primary care consultations between one year to the next (modelling the difference) to understand how much high users contribute to that change? I don't see how the current regression modelling strategy helps achieve the study goal. Also, the current model seems to be almost tautological. High users (defined by number of consultations above a certain threshold) are more likely to use primary care. I know the authors argue that this is expected, but we needed to understand its magnitude. I'm unclear about how the regression strategy gets us there. 2. Interpretation of results: In the results section describing the
--	---

	regression results, the interpretation of the incidence rate ratios is unclear, particularly in the abstract. In my opinion, the authors could provide the incidence rate ratios in parenthesis and explain their interpretation (that high users had about 5 times more primary care consultations than non-users?). Or which was the reference group? 3. Introduction: In my opinion, the structure and content of the introduction could be improved. The first two paragraphs could be combined as both are talking about the increase in primary care workload. I would then quickly move to the main topic of the paper (frequent users). The paragraph on NHS workforce seems a bit disconnected from the rest in its current form. Also, geographic variation seems poorly related to high users (the focus of the paper) and considered as a problem (p.5, lines 58-59). Isn't geographic variation in primary care consultations expected due to different underlying populations with a different age, gender, and multimorbidity composition? Minor points: 4. I would recommend modifying the title so it's more concise and the authors may want to consider not presenting three sample sizes. 5. Don't the authors think that including all consultations may result in an overestimate of the actual primary care workload? Hobbs 2016 article also looked at the duration of consultations. Was that information available to the authors? 6. Page 6, line 24 lacks a closed parenthesis. 7. Is there any reason for the decline in the number of GP practice in CPRD from 740 to 389? 8. A bit more intuition could be provided behind the information that the two measures presented on page 7 convey (r and p), particularly if results have to be useful to clinical practice and accessible to a wide audience. 9. Page 8, lines 22-26 seem to pertain to the methods section rather than the results section.
--	---

VERSION 1 – AUTHOR RESPONSE

Additional reviewer (reviewer # 2):

General comment:

This paper aims to quantify the extent to which frequent attenders drive primary care consultations. The analysis uses an impressively large dataset, CPRD GOLD, with 12m patients and 845 practices. Analyses are conducted at the practice level. The authors motivate the significance of this work by arguing that frequent attenders may have special health needs. The authors also point at the changes in the primary care workforce mix (e.g., more allied health professionals) and care delivery (e.g., more e-consultations). However, the findings from this paper are unable to inform these two areas. Without information on patient and practice characteristics, the significance and usability of this work is critically hindered in my opinion. I provide further comments below. The authors have tried to address the comments from previous reviewers and updated the text accordingly. The new version of the manuscript is clearer but, in my opinion, there are still some major concerns, which I detail below.

Response: We thank the reviewer for the overall assessment of the study and we accept the criticisms, which were already part of the paper in the limitations section. We of course agree with the

limitations of the study, and the next steps needed to shed more light into this research space, but this was a necessary first step in that direction: quantifying the size of the problem.

Major points

1. *The methods section in the abstract should mention the regression analysis. Isn't that the main method to identify the extent to which high users drive primary care use? Under the strengths and limitations section, the regression results are not presented, only the descriptive on the percentage of consultations used by frequent users.*

Response: Thank you for highlighting this omission. The regression analyses were presented in the results section of the abstract, because of the word limit. Similarly, there is a finite number of items for the strength and limitations box, with the focus on strengths and limitations rather than results, and we had decided not to include the regression results since they are generally in agreement with the descriptive analyses, with the latter easier to communicate. We have now removed all results from the strengths and limitations section and have added the regression information to the methods section of the abstract.

2. *In my opinion, the statement of the study goals in the introduction needs to be more focused (p.6, line 3). Particularly "a) different ways to describe consultation distributions in a way that is informative to clinical practice" seems quite general and vague. I think the study would come across as stronger if it was focused on high users.*

Response: Thank you for this comment. We have expanded the description to clarify that there is a focus on frequent attendance – although our aims are unchanged and “different ways to describe consultation distributions in a way that is informative to clinical practice” is still the first aim of the paper, and we feel it is an accurate description of what we present in the methods section.

3. *On page 7, line 48, the study goal is stated as understanding practice characteristics that drive an imbalance consultation distribution. This analysis lacks a clear study hypothesis. Again, I think focusing the paper on high users is preferable.*

Response: We have expanded to clarify the focus is on frequent attendance.

4. *What is the rationale behind the choice of the independent variables in the regression model? Why was deprivation included but not, for example, the percentage of people above 70? or the percentage of women? or practice characteristics (such as available workforce?). Adjusting for at least age is essential to understand the impact of being a "high user", unconfounded by this key factor.*

Response: This is a valid point, and it is an inherent limitation of an “aggregate” practice-level analysis (which were briefly discussed as a response to the reviewer's general comment). Thus we avoid patient aggregates at the practice level and focused on available practice-level covariates. Thus, our model is constrained by data availability at the practice level. At that level of analysis, it is practically impossible to quantify an association between practice location deprivation and consultation imbalance that is adjusted for patient characteristics, thus we did not attempt to. Irrespective of the potentially different characteristics at different deprivation strata, we observe small differences at the practice level, which we have clearly reported as such. The next step would be conducting patient level analyses to examine predictors of frequent attendance, as is explained in the limitations section of this paper.

5. *What was the goodness of fit of the regression model?*

Response: Although goodness of fit measures are widely accepted for linear models, their validity is questionable for non-linear models, hence why we did not originally report them. We have now clarified that pseudo-R² values for the models ranged between 0.048 and 0.109.

6. *Are the measures presented on page 7 (pij and rij) used beyond descriptives? How many regression models were estimated? I'm under the impression that for each frequent user threshold, two regression models were calibrated, one for pij and another for rij? But then I don't understand how the averages for each group in number of consultations come in and how they help understand the distribution of primary care consultations. Please clarify.*

Response: These definitions are only used for descriptive statistics. The outcome used in the negative binomial regressions is the number of consultations, within each group of interest, accounting for the size of the exposure (the number of patients within each group). We have slightly rephrased the regression description in the paper to make this clearer.

7. *In my opinion, the results section should be much more focused on high users and provide a stronger interpretation for the regression results*

Response: We have reviewed the results section to ensure that all findings are clearly presented and interpreted. The descriptive statistics and the regression results report on the levels across the two groups (low/high consultation volume) and their relative differences. As previously discussed, this is a practice-level analyses with certain limitations that cannot be overcome with aggregate data.

8. *The implications for policy do not seem to flow from the study results: "Whilst many of these patients may have comorbidities and may need to be seen regularly, research suggest that they have wider social and psychological needs(17). GPs should be looking at this group of patients more closely to understand who are they and why are they consulting more frequently." The authors use a reference (17) rather than their study findings, which is odd.*

Response: We do not agree with the reviewer here. The findings from our study were presented in a preceding section; so just reiterating them in the 'implication for policy and practice' section would add little value. We believe, it is our duty to draw possible conclusions and recommendations for policy makers in this section after we have placed our findings into the broader research context. Each of our statements accurately reflect our findings, the current state of primary care or other relevant and complementary studies which are cited.

9. *Wouldn't the authors need a model explaining the change in primary care consultations between one year to the next (modelling the difference) to understand how much high users contribute to that change? I don't see how the current regression modelling strategy helps achieve the study goal. Also, the current model seems to be almost tautological. High users (defined by number of consultations above a certain threshold) are more likely to use primary care. I know the authors argue that this is expected, but we needed to understand its magnitude. I'm unclear about how the regression strategy gets us there.*

Response: We thank the reviewer for these queries, and we hope that our answers below clarify our approach. The outcome is the number of consultations in each of the two groups (high/low, as dichotomised by one of the definitions we used), for each practice – practice level aggregates as previously explained. To understand how much high users contribute to that change, we have explained that we have used a model that included a time-group interaction term, the results from which are presented in Figure 3. We accept the tautology criticism, but that is the only way to approach the issue (at least in our opinion) with aggregate data. Necessarily, there will exist a dichotomy, categorising into high and low users. However, the focus of the analysis is not to inform if the top user consult more (as the reviewer implies), of course we know that; but by how much, if that varies across practices and regions, and if that changes over time. The regression analyses, answer these questions (see table 5 and figure 3).

10. *Interpretation of results: In the results section describing the regression results, the interpretation of the incidence rate ratios is unclear, particularly in the abstract. In my opinion, the authors could provide the incidence rate ratios in parenthesis and explain their interpretation (that high users had about 5 times more primary care consultations than non-users?). Or which was the reference group?*

Response: In the abstract we said "Adjusted incidence rate ratios for frequent attenders ranged between 4.992 (95%CI: 4.917, 5.068) for face-to-face consultations with all staff, and 5.603 (95%CI: 5.560, 5.647) for all consultations with GPs, compared to the rest of consulting practice population". We have explained how frequent attenders were defined in the methods section (both in the abstract and the main paper), and we state the reference category. Similarly in the main paper we say "Adjusted incidence rate ratios (IRRs) for the top 10% consultations group ranged between 4.99 (95% CI: 4.92 to 5.07) for face-to-face consultations with all staff, and 5.60 (95% CI: 5.56 to 5.65) for all consultations with GPs, compared to the bottom 90% consultations group". We are of the opinion that the statements are clear and precise, but we have now edited the statement in the abstract to make it easier for non-academic readers to follow, a change that necessarily increased the word count in the abstract.

11 *Introduction: In my opinion, the structure and content of the introduction could be improved. The first two paragraphs could be combined as both are talking about the increase in primary care workload. I would then quickly move to the main topic of the paper (frequent users). The paragraph on NHS workforce seems a bit disconnected from the rest in its current form. Also, geographic variation seems poorly related to high users (the focus of the paper) and considered as a problem (p.5, lines*

58-59). *Isn't geographic variation in primary care consultations expected due to different underlying populations with a different age, gender, and multimorbidity composition?*

Response: We thank the reviewer; we have now revised the introduction accordingly. The first and second paragraphs have been combined into one. The second paragraph (third previously) was introducing frequent attenders, so no action was taken. The third paragraph (previously fourth) introduced workload issues, which are of course integral to any consultations volume discussion, especially in the context of frequent users and the resources utilised to deal with them (and how they contribute to the workload). Geographic variation is also mentioned, as the reviewer highlights, and whatever the underlying cause of any variation (largely it is deprivation and patient characteristics driven) it is still important to quantify "unadjusted" differences, to pinpoint local needs, service planning and inform (ideally) funding allocation.

Minor points

1. *I would recommend modifying the title so it's more concise and the authors may want to consider not presenting three sample sizes.*

Response: We have reduced the length while continuing to conform to the STROBE statement.

2. *Don't the authors think that including all consultations may result in an overestimate of the actual primary care workload? Hobbs 2016 article also looked at the duration of consultations. Was that information available to the authors?*

Response: It depends on the definition of "actual primary care workload". This is the main reason we present all consultation types and face-to-face consultations. Anecdotally, and from the Hobbs paper, it is known that everything except face-to-face consultations is on the increase, and our study is in agreement. Is all other things GPs have to do not "actual" workload? GPs, including members of our authorship team, tend to disagree. Duration is available, we used a similar database as the one used in the Hobbs study. However, the field is notoriously unreliable and missing for most consultations, hence why we decided not to use it.

3. *Page 6, line 24 lacks a closed parenthesis*

Response: Thank you for your careful reading, corrected.

4. *Is there any reason for the decline in the number of GP practice in CPRD from 740 to 389?*

Response: The main reason is the reduction of the market share of the VISION clinical computer system, which was explored in reference 14 (previous work of member of the team). We have now made this clearer in the limitations section.

5. *A bit more intuition could be provided behind the information that the two measures presented on page 7 convey (r and p), particularly if results have to be useful to clinical practice and accessible to a wide audience.?*

Response: Thank you, we have presented the formulae of the measures and described the way for them to be utilised, to quantify the issue within a practice that wishes to quantify the issue we describe. We must admit this is a very difficult comment to act on, since it is quite generic, but we hope it is now addressed.

6. Page 8, lines 22-26 seem to pertain to the methods section rather than the results section.

Response: Thank you, we have moved this sentence to the methods section.

VERSION 2 – REVIEW

REVIEWER	Soley-Bori, Marina King's College London
REVIEW RETURNED	05-Sep-2021

GENERAL COMMENTS	The authors have satisfactorily addressed my comments. I only have two minor suggestions for the authors to consider. -Abstract: The results section would be stronger if the authors started with their main findings (regression analyses). -There is inconsistency in the number of decimal points used
--

throughout the manuscript.

VERSION 2 – AUTHOR RESPONSE

Additional reviewer (reviewer # 2):

General comment:

The authors have satisfactorily addressed my comments. I only have two minor suggestions for the authors to consider.

-Abstract: The results section would be stronger if the authors started with their main findings (regression analyses).

Results: Thank you for this suggestion, but we would prefer to leave the abstract as is. We do not consider the practice-level regressions to be the main findings and it is common practice to start with the descriptive results.

-There is inconsistency in the number of decimal points used throughout the manuscript.

Results: Thank you. There is, but it is deliberate because we report very different measures, for example, percentages for which more than one decimal does not make much sense, and regression estimates where two decimals are needed to inform on the confidence interval. Thus, we would prefer to make no changes, but we will be happy to discuss this with the production team.